# Insights into the Multifaceted Roles of Thioredoxin-1 System: Exploring Knockout Murine Models

**DOI:** 10.3390/biology13030180

**Published:** 2024-03-12

**Authors:** Tetiana Shcholok, Eftekhar Eftekharpour

**Affiliations:** Department of Physiology and Pathophysiology, University of Manitoba, 631-BMSB, 745 Bannatyne Ave, Winnipeg, MB R3E 0J9, Canada; shcholo1@myumanitoba.ca

**Keywords:** redox balance, thiol antioxidants, transgenic models, brain, heart, liver, pancreas, glutaredoxin, thioredoxin reductase

## Abstract

**Simple Summary:**

This article reviews the biological importance of the Thioredoxin-1 system in different cells and organs. Redox balance, defined by an equilibrium between oxidized and reduced molecules, is crucial for proper cellular functions in health. Disruption of this balance may lead to cell death. Oxidation and reduction of cysteine residues in key signaling molecules changes their structural conformation and function. Thioredoxin-1 is a key regulatory protein that is used as a buffer to maintain the proteins in their optimal shape. In this review, we specifically focus on transgenic animal models related to the Thioredoxin-1 system. This review aims to showcase the specific roles of the Thioredoxin-1 system in maintaining balance in various organs and cell types.

**Abstract:**

Redox balance is increasingly identified as a major player in cellular signaling. A fundamentally simple reaction of oxidation and reduction of cysteine residues in cellular proteins is the central concept in this complex regulatory mode of protein function. Oxidation of key cysteine residues occurs at the physiological levels of reactive oxygen species (ROS), but they are reduced by a supply of thiol antioxidant molecules including glutathione, glutaredoxin, and thioredoxin. While these molecules show complex compensatory roles in experimental conditions, transgenic animal models provide a comprehensive picture to pinpoint the role of each antioxidant. In this review, we have specifically focused on the available literature on thioredoxin-1 system transgenic models that include thioredoxin and thioredoxin reductase proteins. As the identification of thioredoxin protein targets is technically challenging, the true contribution of this system in maintaining cellular balance remains unidentified, including the role of this system in the brain.

## 1. Introduction

The importance of cellular redox status in cell physiology is increasingly identified. Reactive oxygen species (ROS), such as superoxide anion and H_2_O_2_, are critical secondary messengers in cells during health and disease. Their messenger role is achieved by the oxidation of key cysteine residues in different signaling proteins, a highly efficient way to regulate protein activity. These oxidized residues must be quickly reduced to keep the cell in an optimal state. The reducing equivalents are provided by nicotinamide adenine dinucleotide phosphate (NADPH). These are transferred to glutathione (GSH)/glutaredoxin and thioredoxin (Trx) systems through GSH-reductase (GR) and Trx-reductase (TrxR) enzymes. Despite seemingly overlapping functions of GSH and Trx systems, their mode of action is quite unique. Reduction of oxidized substrates via GSH requires the formation of a glutathione adduct. These adducts are stable and can be detected using several mass spectrophotometry methods; hence, the importance of GSH in neurobiology is well documented. Trx reduces its substrate through the formation of a transient mixed disulfide intermediate compound, followed by a fast thiol–disulfide exchange reaction with oxidized cysteines in the target protein. This results in the oxidation of Trx and the reduction of its substrate. Since there are no stable Trx adducts in this process, the identification of Trx targets is a very challenging task, therefore the extent of Trx contribution to cell signaling systems remains vastly unknown. Early attempts in creating a whole-body knockout (constitutive) of the Trx system resulted in the death of embryos before gastrulation [1,2]; therefore, several cell-specific and organ-specific knockout mouse models have been generated to better understand this protein. These models generally have a shortened life span but have provided a glimpse of Trx-1′s contribution to cell biology.

This review further explores the intricate mechanisms and physiological implications of the Trx-1 system. It provides an overview of the Trx system’s enzymes, focusing on the isoforms of mammalian Trx and their distinct functionalities. Additionally, it discusses the roles of the Trx-1 system in cellular physiology, highlighting its significance in maintaining redox homeostasis. Furthermore, this review offers a comprehensive analysis of murine Trx-1 system knockout models, elucidating the molecular, morphological, and functional characteristics of these models in various organs. Through a comparative analysis of organ-specific and cell-specific Trx-1 knockout models, this review aims to dissect the common functions of the Trx-1 system across different tissues, as well as those specific to particular cell types or organs. Together, these discussions provide valuable insights into the intricate mechanisms and physiological implications of the Trx-1 system, offering a comprehensive resource for researchers exploring its tissue- and cell-type-specific roles.

## 2. The Trx System’s Enzymes

Thioredoxins are small antioxidant proteins with a conserved active site (Cys32-Pro-Gly-Cys35) in all organisms and work as general oxidoreductases. These proteins react with a wide range of proteins and control their stability and function. The name “thioredoxin” originated from the molecular mechanism that underlies their redox function. Two readily accessible electrons on cysteine residues at Trx active sites (Cys32-Gly-Pro-Cys35) are used for their redox activity (Figure 1). Trx-1 has a low affinity for H_2_O_2_ [3]; however, it is an important oxidoreductase protein and can directly reduce the oxidized proteins through a kiss-and-run reaction. During this reaction, Trx first donates an electron from its Cys32 to the oxidized substrate, forming a transient disulfide bond with the target. The second electron at Cys35 is then donated to the oxidized substrate, fully reducing it [3]. This results in oxidation of Trx by forming an intracellular disulfide bond between Cys35 and Cys32. The oxidized Trx must be reduced by TrxR using two electrons from NADPH, before resuming the redox regulation activity. The reduced Trx can also act as an electron donor for peroxiredoxins (Prx). This Trx-dependent peroxidases are responsible for peroxide scavenging and have been reviewed previously [4]. Trx can be inactivated by binding to its natural endogenous inhibitor, thioredoxin-interacting protein (TXNIP), which is also known as vitamin D3 Upregulated protein (VDUP). TXNIP binding to Trx results in exacerbating oxidative stress [5], as shown in type-1 diabetes, resulting in β-cell death [6].

There are several other proteins in eukaryotes that are evolutionally related to Trx that share structural similarities with Trx. These are mostly members of a group of proteins known as protein disulfide isomerase, which are involved in proper protein folding in the endoplasmic reticulum. These proteins have been reviewed recently [7,8] and will not be discussed here.

### 2.1. Isoforms of Mammalian Trx

To date, three Trx isoforms have been identified in mammals: Trx-1, Trx-2, and Trx-3 (Table 1). While Trx-1 and Trx-2 are encoded by separate genes, Txn-1 and Txn-2, while Trx-3 results from agenetic duplication of the Txn1 gene [9]. Despite their relatively similar structures with a conserved disulfide active site sequence (Trp-Cys-Gly-Pro-Cys) [10], their distinct subcellular and tissue-specific localizations determine their specific functions. Thioredoxin isoforms can also be categorized into two subgroups: (a) those ubiquitously expressed in all cell types (Trx-1 and Trx-2) and (b) testes-specific (Trx-3) [11]. Trx-3 is predominantly localized in the Golgi apparatus of mammalian spermatids and spermatocytes [9]. Interestingly, the expression of Trx-3 is notably elevated in defective spermatozoa, making this isoform a promising diagnostic marker for detection of aberrant spermatogenesis and infertility in males [9].

Trx-1 and Trx-2 are the most researched isoforms of this family. We will first briefly discuss Trx-2 and then we will focus primarily on Trx-1. Trx-2 is predominantly localized in mitochondria [12]. Mitochondria, as the primary producers of reactive oxygen species (ROS) in mammalian cells, heavily rely on redox control through mitochondrial antioxidant systems. Trx-2′s general function in mitochondria involves maintaining membrane potential by reducing the protein disulfides, which impacts mitochondrial function. Downregulation of Trx-2 leads to decreased ATP/ADP ratio, reduced oxygen consumption, elevated lactate production, and activation of caspase 3 and caspase 7 [13]. Trx-2 also plays a role in tumor necrosis alpha/apoptosis signal-regulating kinase-1 (TNF-α/ASK-1)-stimulated release of cytochrome C during apoptosis [14]. Thus, Trx-2 binds to Cys-30 in the N-terminal domain of ASK1 and prevents its activation. This was demonstrated by Zhang et al. with the overexpression of Trx-2 in human umbilical vein endothelial cells, which prevented ASK1-induced apoptosis without significant effects on ASK1-mediated c-Jun N-terminal kinase activation [15]. Other reports confirm that Trx-2 depletion in vitro has a pro-apoptotic effect [14,16]. Trx-2 expression is higher in tissues with heavy metabolic rate, e.g., stomach, testis, ovary, liver, brain, heart, and adrenal gland [16]. Furthermore, Trx-2 plays an important role in embryogenesis, as the constitutional Trx-2 knockout in mice is embryonically lethal [17]. In this work, mutant animals displayed defects in neural tube closure and dramatic activation of apoptosis.

Trx-1 is found mainly in the cytoplasm of all cells, but it is also translocated to the nucleus in response to oxidative or nitrosative stress [18]. Some reports indicate its secretion to the extracellular space [19,20]. Both Trx-1 and its truncated form (Trx-80) are secreted by immune cells (e.g., monocytes, lymphocytes, and neutrophils), playing a role in cell-to-cell communication and facilitation of chemotaxis [21]. This mechanism is especially important in infection, inflammation, and neoplastic changes in the hematopoietic system [22]. These features make the Trx-1 protein a unique member of cellular redox system.

Trx-1 was first isolated from *E. coli* in 1964 [23] and was sequenced by the late Dr. Arne Holmgren [24]. The contributions of Dr. Holmgren have played a major part in shaping the existing knowledge of Trx. Prior to its discovey, Trx-1 had been identified under various names as a hydrogen donor facilitating the reduction of methionine sulfoxide and sulfate with NADPH in yeast [11,21]. The Trx-1 system is conserved across all botanic and animal species [23], and extensive structural data exist with examples of crystal structure for yeasts (*Saccharomyces cerevisiae*, *Schizosaccharomyces pombe*) plants (*Arabidopsis thaliana*), bacteria (*E. coli*, *M. tuberculosis*), protozoa (*T. vaginalis*, *P. falciparum*, *Dictyostelium discoideum*), nematodes (*Caenorhabditis elegans*), and mammals (*Rattus norvegicus*, *Mus musculus*, *Bos taurus*, *Equus caballus*, *Ovis aries*, *Sus scrofa*, *Homo sapiens*) [21,25]. The active site of the Trx-1 molecule (Trp-Cys-Gly-Pro-Cys) [11] is conserved across all the species described to date, indicating its structural importance for protein function [23].

Interestingly, mammalian Trx-1 has a slightly different structure compared to other species [11]. In addition to Cys32 and Cys35 located in its active site, it contains three other cysteine residues (Cys62, Cys69, and Cys73) [11]. To date, the importance of these additional cysteines remains controversial; however, it has been shown that oxidation of cysteines located outside of the active site leads to loss of Trx-1 enzymatic activity [26]. While both Cys62 and Cys69 undergo only S-nitrosylation [27,28,29,30], Cys73 is a multimodification site that can undergo dimerization [31], nitrosylation [32,33], gluthathionylation [33], or 4-hydroxy-2-nonenal modification [34]. In comparison, Trx-2 has only one Cys residue outside its active site. Therefore, it is less susceptible to stress-induced post-translational modifications [35].

### 2.2. Roles of the Trx-1 System

Trx-1 is a multifaceted protein with multiple roles in cellular homeostasis. While its disulfide reductase activity is well known, Trx-1 can also modify its substrates by transnitrosylation or denitrosylation of specific proteins; therefore, it is involved in numerous vital cellular processes, such as cell proliferation, differentiation, migration, apoptosis, autophagy, inflammation, and other metabolic activities [36]. Trx-1-mediated reactions are fast, and the “Trx-1target” intermediate complexes are not easily detectable. Hence, the importance of this protein in regulation of protein functions resembles the role of an invisible conductor for the protein orchestra. The known contributions of Trx-1 in various cellular processes are summarized in Table 2.

The reduction of oxidized Trx-1 by TrxR-1 is essential for ensuring Trx-1 availability because TrxR-1 is the only reducing enzyme for Trx-1. It is generally expected that mutations affecting TrxR-1 activity will have a similar outcome as the Trx-1-null mutation [65]. In contrast, Du et al. reported that in some cellular models, cell viability is not affected after the pharmacological inhibition of TrxR-1, as glutathione and glutaredoxin systems are able to functionally replace TrxR-1 reducing action of Trx-1 in HeLa cells [66]. However, our group has reported that the inhibition of TrxR-1 [67] or Trx-1 [44] significantly affected cell survival in neuroblastoma cells. This may reflect the difference between the cellular models used in these studies.

## 3. Review of Murine Trx-1 System’s Knockout Models

In this section, we provide an overview of the murine Trx-1 system’s knockout models generated by using four distinct genetic modification techniques. These techniques include constitutive gene knockout, inducible gene knockout, catalytically inactive gene knock-in, and organ-specific gene knockout.

Constitutive Trx-1 and TrxR-1 knockout models involve the genetic ablation of Trx-1 and TrxR-1 genes in all tissues and cells throughout the organism’s development. These models are valuable for elucidating the fundamental roles of the Trx-1 system in embryonic development, viability, and overall organismal physiology.

General inducible Trx-1 knockout and catalytically inactive Trx-1 models offer a dynamic approach to study the temporal and tissue-specific effects of Trx-1 loss. By allowing researchers to conditionally delete or inhibit Trx-1 expression in a spatially and temporally controlled manner, these models enable the investigation of Trx-1′s role in specific physiological processes and pathological conditions. Additionally, catalytically inactive Trx-1 models, which express mutant forms of Trx-1 that competitively inhibit endogenous Trx-1 function, provide insights into the molecular mechanisms underlying Trx-1-mediated redox signaling and cellular responses. Organ-specific Trx-1/TrxR-1 knockout models are designed to specifically target Trx-1 or TrxR-1 expression in particular organs or cell types, allowing for the assessment of tissue-specific functions of the Trx-1 system. These models are instrumental in delineating the roles of Trx-1 in organ-specific physiology, pathophysiology, and disease progression. Throughout this review, we will discuss the advantages and limitations of each category of knockout models, outlining their practical applications and justifying the reasons for their utilization. For readers interested in delving into the technical aspects of murine model generation and specific genetic manipulations, as well as exploring the nuances of their application, their advantages, and their disadvantages, we recommend consulting our recent review paper [68].

### 3.1. Constitutive Trx-1 and TrxR-1 Knockouts

Global inborn Trx-1 deletion is reported to be embryonically lethal [1]. While the heterozygote mice appear to be normal and fertile, embryos with a complete Trx-1 knockout show rapid growth retardation at the stage of a blastocyst (approximately 3.5 days of gestation). Trx-1-null embryos had significant deficits in trophoblast formation. Ex vivo experiments revealed defective hatching from zona pellucida which was due to significant loss of proliferative capacity. The authors linked these findings to the role of Trx-1 in DNA synthesis, previously shown in E. coli [1]. These findings emphasized the indispensable biological role of Trx-1 and evoked interest in the investigation of underlying pathological mechanisms associated with Trx-1 deficiency in mammals. The importance of the Trx system in embryogenesis was further confirmed by Jakupoglu et al., who used the Cre-LocP recombination technique [68] for the excision of exon 15 in TrxR-1 [69]. Similar to Trx-1 general knockouts, these animals exhibited early embryonic lethality. However, the mean survival of these embryos was longer than in Trx-1 knockouts with an average resorption of knockout embryos from 9.5 to 10.5 gestational days [69]. TrxR-1-deficient embryos displayed growth retardation with a significantly shortened anterior–posterior axis, failure of neural tube closure, and other signs of impaired somatogenesis. Interestingly, despite dramatic dysmorphogenesis in most vital organs, heart development showed no significant deviations from controls. Histological examination revealed no apparent signs of apoptosis activation in TrxR-1-deficient embryos, while cell proliferation was strikingly reduced in all cell types except for cardiomyocytes.

One possible interpretation is that alternative redox regulatory pathways may partially compensate for the loss of TrxR-1 activity in the developing heart, thereby maintaining essential redox homeostasis and supporting normal cardiac morphogenesis. The reduction of oxidized Trx-1 via GR and glutaredoxin as an alternate for TrxR-1 has been reported [66]. Furthermore, this finding underscores the complexity of organogenesis and highlights the need for further investigation into the specific roles of redox regulators in different tissues and developmental contexts. Future studies utilizing tissue-specific knockout models and advanced molecular techniques may provide deeper insights into the intricate interplay between redox signaling pathways and organ development, ultimately enhancing our understanding of embryogenesis and potentially informing therapeutic strategies for congenital heart defects and other developmental disorders. Given the described association between the loss of TrxR-1 and impaired cell differentiation, future investigations may focus on exploring signaling pathways known to regulate differentiation processes and examine their modulation in TrxR-1-deficient cells or tissues (e.g., Notch, Wnt, BMP (bone morphogenetic protein), and Hedgehog signaling, which are known to play crucial roles in cell fate determination and differentiation during embryonic development) [69].

Bondareva et al. further elaborated on the characterization of the TrxR-1-null phenotype [2]. This group generated TrxR-1-deficient mice via excising exons 1 and 2 of TrxR-1 gene. Consequently, the mutant protein lacking both N-terminal active site cysteines (Cys59 and Cys64) was functionally null. These TrxR-1-null embryos showed delayed development at an earlier age (8.5 embryonic day) compared to those described by Jakupoglu [69]. A pathohistological assessment revealed that embryonic development was interrupted at the pre-streak stage; therefore, mesoderm and ectoderm layers did not form in these embryos. Authors associate the reduction in embryo size with disruptions of differentiation rather than proliferation as mutant embryos were capable of generating several thousands of cells with no apparent change in cellular size, as indicated by the intracellular coefficient that is used for the assessment of cell size. Of note, several markers of early differentiation were found in TrxR-1 knockout embryos (e.g., Cripto, Fgf8, Snail1, Mash2, and Pl1), suggesting partial differentiation in these embryos [2].

### 3.2. General Inducible Trx-1 Knockout and Catalytically Inactive Trx-1

To avoid embryonic lethality in Trx-1-deficient animals. Jabbar et al. were the first to employ a tamoxifen-inducible model of general Trx-1 knockout [70]. Upon the induction of Trx-1 depletion, these animals experienced significant deterioration in their health condition and could survive only up to 30 days post-injection of tamoxifen with a mean survival of 15 days. The group reported robust changes in the spleen, with an approximately 30% decrease in wet mass and a 50% decrease in splenocyte count compared to the control. Flow cytometry revealed a significantly increased number of apoptotic spleen cells. A histopathologic assessment revealed no signs of inflammation, necrosis, cell degeneration, or embolism in other organs (e.g., brain, heart, lung, liver, adrenal glands, or kidney). However, some changes in the morphology of the gastrointestinal tract (disorganized intestinal villi and inflammatory cell infiltration) were observed [70].

Using the catalytically inactive form of Trx-1 is an alternative approach to disrupt the activity of the wildtype Trx-1. This was achieved using site-directed mutagenesis in human Trx-1, and the active Cys 32 and 35 residues were mutated to serine, leading to a loss of Trx-1 activity. The transgene vector was injected into the pronuclei of fertilized eggs. This strategy introduces an overexpression of mutant/inactive Trx-1, which competes with the wildtype. This approach may offer a direct and targeted means of modulating Trx-1 function without the need for genetic ablation, enabling the investigation of acute and reversible effects of Trx-1 inhibition on cellular processes. Furthermore, the catalytically inactive approach provides flexibility in modulating Trx-1 activity in a spatially and temporally controlled manner, depending on the choice of promoter driving the expression of the inactive Trx-1 construct. This allows for the selective manipulation of Trx-1 function in specific cell types or tissues of interest, facilitating the study of the tissue-specific roles of Trx-1 in various physiological and pathological contexts. When compared to the tamoxifen-inducible approach, catalytically inactive models may be more suitable for immediate and sustained gene inhibition, while tamoxifen-inducible models may be preferable for the finely tuned, reversible modulation of gene expression [68].

These animals expressed high levels of non-functional human Trx-1 and low levels of wildtype mouse Trx-1 [70]. Although it is not specifically stated in this article, the animals were seemingly able to survive under normal conditions. This study mainly focused on the protective role of Trx-1 in a hyperoxic lung injury model. When exposed to hyperoxia, 100% of the mice died within 72 h of recovery period. Histopathologic assessments revealed increased alveolar damage with focal hemorrhages in alveoli and hyaline membrane deposits. On a cellular level, a decrease in ATP production and non-mitochondrial respiration was observed in both normoxic and hyperoxic conditions. Despite an increase in the expression of cell cycle regulator and DNA damage sensor p53 in non-functional Trx-1 animals in both normoxic and hyperoxic conditions, apoptosis was not found to be the main mechanism of cell death in this animals after exposure to toxic concentrations of oxygen. Moreover, neither of the pro-apoptotic markers (cleaved caspase 3, cleaved PARP, Bax, BCL2, JNK, pASK, etc.) analyzed in this study showed a significant increase compared to the controls. Consequently, the authors suggested necrosis to be the underlying mechanism of severe and rapid lung damage after a hyperoxic injury. A transgenic mouse overexpressing wildtype Trx-1 in this study was used to confirm the involvement of Trx-1 in protection against hyperoxia [71].

### 3.3. Organ-Specific Trx-1/TrxR-1 Knockouts

#### 3.3.1. Heart-Specific Depletion of Trx-1 Activity

The role of Trx-1 in cardiac muscle health has been previously shown in several in vitro studies and reviewed previously [72]. A model of cardiac-specific Trx-1 functional depletion was first generated by Yamamoto et al. [73]. A redox inactive isoform of human-Trx-1 (hTrx-1) was expressed in cardiomyocytes using α-myosin heavy-chain Cre mice strain. In this model, the disulfide oxidoreductase activity of endogenous Trx-1 was suppressed by mutant hTrx-1. When kept in basal conditions, these mice developed a concentric form of cardiac hypertrophy with a significantly enlarged septum and left ventricle wall as assessed by histology and echocardiography. Despite these morphological abnormalities, the mice did not show systolic or diastolic dysfunction. The authors linked the observed changes in myocardium to severe oxidative distress caused by loss of Trx-1 function. The cardiomyocytes in these animals displayed signs of oxidative stress-induced DNA damage (evidenced via 8-OHdG staining) and enhanced lipid peroxidation (which was shown via 4HAE staining). There were no signs of altered cell proliferation in these mice, suggesting that cardiomyocyte hypertrophy—not cell proliferation—was the sole contributor to the enlargement of the cardiac muscle. Mechanistically, the loss of Trx-1 function resulted in the activation of the ERK pathway, a well-known player in concentric heart hypertrophy and a protector against pro-apoptotic stimuli [74].

One of the latest advances in dissecting the role of Trx-1 in heart metabolism and function was the generation of the Trx-1 heart-specific knockout model by Oka et al. This group was the first to report a heart-specific loss of function model [46]. Cardiomyocyte-specific Trx-1 deletion was achieved by breeding Trx-1fl/fl animal with Myh6-Cre (myosin heavy-chain 6 Cre) mouse. These animals had dramatically shortened lifespans (median survival 25.5 days) due to significant functional deficits in heart function. Cardiac dilatation hypertrophy with decreased left ventricle ejection fraction in these mice resulted in chronic heart failure. At the cellular level, Trx-1-deficient cardiomyocytes showed a two-fold increase in size compared to controls with a large proportion of apoptotic cells as demonstrated via the TUNEL assay. Furthermore, a nearly two-fold activation of caspase 3 levels confirmed the domination of pro-apoptotic response in Trx-1-null heart muscle cells. In addition to the involvement of Trx-1 in regulation of cell proliferation that was observed in other studies, Trx-1′s protective function has also been linked to regulation of autophagy. This highly regulated recycling program is responsible for removing damaged proteins and organelles. Accordingly, the interruption of autophagy in Trx-1 heart-specific knockouts was shown through oxidation of mTOR, the master regulator of autophagy in cells. The activation of autophagy is assessed by increased levels of p62 and the LC3II/LC3I ratio. Authors also showed that the lack of Trx-1 resulted in the oxidation of mTOR at Cys 1483; however, mTOR inactivation did not result in changes in p62 or LC I/II levels. Surprisingly, mTOR inhibition in this model contributed to dysregulation of metabolic genes and mitochondria dysfunction, which are critical for the health of highly specialized cell types with enhanced metabolic turnover, such as cardiomyocytes [75]. Similar to other cardiac-specific gene knockouts that impair mitochondria health, the disruption of mitochondria metabolism is viewed as the dominant cause of cardiomyopathy and heart failure in Trx-1-deficient mice [46].

In summary, while both models of Trx-1 depletion and overexpression of the non-functional Trx-1 protein demonstrate cardiac hypertrophy in response to Trx-1 disruption, the Trx-1 Myh6-Cre heart-specific knockout model exhibits more severe functional deficits, which is illustrated by the development of chronic heart failure and decreased lifespan. Additionally, whereas Yamamoto et al. demonstrated the activation of antiapoptotic mechanisms in Trx-1-deficient cardiomyocytes, Oka et al. attributed functional deficits in Trx-1-depleted cardiomyocytes to the significant activation of apoptosis. This may also reflect the fact that a low-level expression of wildtype Trx-1 in dominant negative overexpression models is sufficient to prevent apoptosis activation in normal conditions. Overall, these distinct differences between the two models highlight the importance of the methodology used when interpreting experimental outcomes.

#### 3.3.2. Liver-Specific Trx-1 and TrxR-1 Knockouts

The metabolism of hepatocytes is associated with an excessive production of ROS; therefore, these cells heavily rely on cellular antioxidants [76]. Interestingly, liver-specific downregulation of Trx-1 and TrxR-1 was not lethal in mice [65]. Despite some lethality (15%), both sexes of adult mice were fertile. The adult mice had larger livers due to hepatocyte proliferation. The hepatocytes maintained their redox balance through de novo GSH synthesis using methionine; however, as expected, the hepatocytes were not able to reduce the oxidized GSH (GSSG). When elaborating on potential rescue mechanisms in this model, Prigge et al. suggested that increased cell death was counterbalanced by elevated rates of hepatocyte proliferation, aligning with the increased cell turnover rates, therefore preserving the homeostasis of the cell population. The group then attempted a triple liver-specific knockout mice model of null Trx-1/TrxR-1/GR [65]. Trx-1 deletion in hepatocytes did not cause overall liver pathology and it did not decrease the lifespan of these animals. Despite the absence of obvious morphological changes, an ultrastructural analysis of Trx-1-null hepatocytes showed a significant decrease in mitochondria cross-sectional area compared to controls, with mitochondrial networks composed of thinner branches. However, this did not result in mitochondrial metabolism dysfunction. Another finding was the increased size of nuclei in Trx-1-deficient hepatocytes. However, due to a slight but not significant increase in hepatocyte volume, the nucleus/cytoplasm ratio was not altered [65]. A comparison of serum from wildtypes with mice with TrxR-1/GR-null livers showed evidence of chronic liver damage, although all hepatic functions were reported to be unaffected.

Another study on TrxR-1 deficiency in hepatocytes used Cre-LoxP recombinase under the control of an albumin promoter [77]. For this purpose, they bred TrxR-1fl/fl and Alb-Cre animals. The offsprings were fully viable with a mean survivance of more than 1 year. Despite anticipated disruption in the redox system, this group did not detect any evidence of oxidative distress in TrxR-1-deficient livers. A transcriptome assessment showed evidence of increased oxidative stress in this model as 21 out of 56 upregulated genes contained antioxidant response elements (AREs) and belonged to the Nrf2 pathway. The proposed activation of the Nrf2 pathway was supported by the upregulation of nuclear Nrf2 in TrxR-1-depleted hepatocytes, indicating its increased cytoplasmic to nuclear translocation. Canonically, Nrf2 activation is triggered by oxidative stress. The possibility of the Nrf2 pathway’s upregulation in the absence of oxidative damage and the role of the Trx-1 system in this process remains unexplored. Overall, a normal lifespan and morphology, as well as an absence of oxidative stress markers, indicate the presence of robust compensatory mechanisms in TrxR-1-depleted hepatocytes [77].

Another study by this group further examined the role of TrxR-1 in liver metabolism in this model. Here, they investigated the impact of TrxR-1 deficiency on the replicative potential of hepatocytes during development and regeneration [78]. This study demonstrated no significant difference in liver growth rate and the number of proliferating hepatocytes, as evidenced via PCNA and phosphohistone H3 staining, and BrdU incorporation. These findings led authors to the conclusion that Trx-R1 is not indispensable for DNA replication and normal growth in the liver [78].

Further studies by this group examined the impact of liver-specific TrxR-1 ablation on hepatic lipogenesis, glycogen synthesis, and detoxification [79]. They noted that TrxR-1-deficient livers exhibit repression of lipogenic genes, which results in glycogen accumulation in periportal hepatocytes. The downregulation of lipogenic genes and the subsequent accumulation of glycogen in TrxR-1-deficient livers suggest a shift in energy metabolism and storage within hepatocytes. Additionally, as a mechanism compensatory to TrxR-1 deletion, mutant hepatocytes upregulated the machinery for glutathione biosynthesis (glutathionylation and glucuronidation), perhaps indicating an enhanced detoxification capacity [79]. This heightened detoxification ability may confer a protective advantage against xenobiotic-induced liver injury, as evidenced by the resistance to acetaminophen-induced pathology observed in TrxR-1-null livers. Therefore, these modifications in hepatocyte metabolism resulted in the preconditioning of mutant hepatocytes to even more robust elimination of xenobiotics compared to wildtype littermates [79].

The changes in hepatic function and metabolism have broad implications for overall liver health and susceptibility to environmental toxins and drug-induced liver injury [79]. The altered metabolic profile and enhanced detoxification capacity may contribute to the resilience of TrxR-1-deficient livers against certain pathological conditions, but further studies are warranted to fully elucidate the long-term consequences and potential adaptive responses associated with TrxR-1 deficiency in hepatic physiology. Additionally, the compensatory mechanisms that are upregulated after genetic manipulation studies must be examined in detail before any drawing any conclusions on the differential importance of thiol-reducing systems.

A recent publication from this group elucidates the role of TrxR-1 during acute cholestatic liver injury [80]. In this study, they subjected mice with liver-specific deletion of TrxR-1 to bile duct ligation surgery. Unlike the wildtype, this approach did not cause any histological signs of necrosis or enhanced fibrogenesis in TrxR-1-depleted livers. The TrxR-1-depleted hepatocytes also had a significantly lower expression of the NLRP3 inflammasome complex along with a less prominent expression of proinflammatory cytokines. These findings highlighted the role of TrxR-1 in the regulation of pro-inflammatory responses in acute cholestasis [80].

#### 3.3.3. Pancreatic β-Cell-Specific TrxR-1 Knockout

Oxidative distress is considered a major mediator of pancreatic β-cell damage [81]. ROS overproduction has been documented in pancreatic tissue derived from diabetic patients and animal models [82,83,84,85,86,87]. There is some supporting literature that links this susceptibility to lower antioxidant proteins in β-cell when compared to other tissue types, e.g., liver or kidney [88,89]. High glucose levels may be the initiation factor in the induction of oxidative stress, as shown previously. The thioredoxin-interacting protein (TXNIP) is upregulated in these conditions, binding to Trx-1, and this may lead to enhanced oxidative stress-mediated cell death in β-cells [90]. Further support for the importance of the Trx-1 system in these cells comes from in vitro reports in which H_2_O_2_ sensitivity of β-cells is attributed to Trx-1 and peroxiredoxin systems [91]. Apoptotic cell death was enhanced in this model after the genetic depletion of TrxR-1 [91].

Importantly, there was no evidence of any secondary mechanism to protect β-cells from such oxidative damage. Stancill et al. questioned whether these findings could be replicated in in vitro conditions [92]. In their study, Insulin-Cre (Ins-Cre) mice were crossed with TrxR-1fl/fl to generate β-a cell-specific TrxR-1 knockout. Functionally, there was no difference in glucose tolerance after a 4 h fasting period or in plasma insulin levels 15 min after glucose challenge in mutants of both sexes compared to controls. Despite this fact, male knockouts demonstrated slightly but significantly increased blood glucose after 4 h of fasting; in females, their glucose concentrations were similar to the controls. An ex vivo assessment of pancreatic islands derived from the knockouts showed a significant reduction of glucose-stimulated and membrane depolarization-stimulated insulin secretion, highlighting the importance of TrxR-1 in β-cell function. Intriguingly, the reduction of β-cell function in this model was not due to apoptosis but rather because of disturbances in the maturation of TrxR-1-deficient β-cells. The identification of factors involved in β-cell maturation involved in the regulation of glucose sensing (e.g., Mafa, Pdx1, Ucn3, Slc2a2, and Gpd2) may explain the observed deficits [92]. Understanding how these factors coordinate the β-cell maturation process is crucial for unravelling the underlying molecular pathways, including insulin secretion and glucose sensing. In terms of potential molecular mechanisms or factors involved in β-cell maturation and their relation to glucose sensing, several hypotheses can be considered. For instance, Mafa and Pdx1 are transcription factors known to regulate the expression of genes involved in insulin biosynthesis and secretion [92]. Dysregulation of these factors may lead to alterations in the expression of glucose transporters (e.g., Slc2a2) or enzymes involved in glucose metabolism, thereby affecting the β-cell’s ability to sense and respond to changes in glucose levels [92]. Similarly, Ucn3 has been implicated in insulin secretion and glucose homeostasis, suggesting its potential involvement in β-cell maturation and function. Additionally, factors such as Gpd2, which is involved in glycerol metabolism, may play a role in β-cell development and function by modulating cellular energetics and redox balance [92]. Overall, the identification of factors affecting β-cell maturation via TrxR-1 deletion and their impact on glucose sensing can provide critical insights into the molecular mechanisms underlying β-cell function. Further research into the identification of these regulatory networks may offer novel therapeutic targets for the treatment of β-cell dysfunction.

#### 3.3.4. T-Cell-Specific TrxR-1 Knockout

Previous studies have demonstrated the role of Txnip/Trx-1 system in the regulation of glucose metabolism [93]. TXNIP-deficient mice display hyperglycemia, hyperinsulinemia, and liver steatosis [93]. TXNIP is reported to be significantly reduced in activated T-cells, while Trx-1 and TrxR-1 are upregulated. This mechanism is crucial for DNA synthesis during metabolic reprogramming of T-cells [94]. Additionally, activated T-cells require high amounts of glucose and amino acids and therefore upregulate glycolysis and glutaminolysis [94,95].

The reported reduction of TXNIP in activated T-cells, coupled with the upregulation of Trx-1 and TrxR-1, suggests the existence of potential compensatory mechanisms in the absence of TrxR-1 in T-cells. One possible such mechanism could involve the activation of alternative redox regulatory pathways to maintain cellular redox balance in the absence of TrxR-1. For example, other antioxidant systems such as the GSH/glutaredoxin system may ecome upregulated to counteract oxidative stress and preserve cell viability [96]. Additionally, enzymes involved in NADPH regeneration, such as glucose-6-phosphate dehydrogenase (G6PD) or isocitrate dehydrogenase (IDH), can be upregulated to ensure that there is an adequate supply of reducing equivalents for essential biosynthetic processes. Furthermore, T-cells may adapt their metabolic profile to compensate for the loss of TrxR-1 by enhancing alternative metabolic pathways. For instance, T-cells may shift toward increased reliance on fatty acid oxidation or mitochondrial respiration to meet their energy demands and maintain cellular function in the absence of TrxR-1-mediated redox regulation. Additionally, T-cells can possibly employ mechanisms to enhance nucleotide biosynthesis and DNA synthesis in the absence of TrxR-1. This could involve the upregulation of rescue pathways for nucleotide synthesis, or increased expression of enzymes involved in de novo purine and pyrimidine biosynthesis to overcome deficits in ribonucleotide reductase activity [96]. Overall, the compensatory mechanisms in the absence of TrxR-1 in T-cells likely involve a complex interplay of redox regulatory pathways, metabolic adaptations, and alterations in nucleotide biosynthesis to maintain cellular viability and function under oxidative stress conditions. Further research is needed to elucidate the specific mechanisms underlying these compensatory responses and their impact on T-cell-mediated immunity.

To investigate the role of the Trx-1/TrxR-1 system in T-cell mediated immunity, Muri et al. developed a model of T-cell specific TrxR-1 knockout [96]. They crossed TrxR-1fl/fl with a CD4-Cre animal, which drives Cre-recombinase expression to CD4 and CD8 double-positive thymocytes. Consistent with the role of the Trx-1/TrxR-1 system in providing reducing equivalents for ribonucleotide reductase [97], TrxR-1-depleted thymocytes had improper purine and pyrimidine synthesis. Muri et al. also reported an accumulation of metabolites in glutamate and aspartate metabolic pathways that were identified as another sign of impaired nucleotide biosynthesis. Morphologically, TrxR-1-deficient thymuses were significantly reduced in size. This was typically associated with a dramatic decrease in both the number and size of thymocytes. A further investigation failed to detect any signs of apoptosis activation. In contrast, the decreased expression of the CD4 and CD8 T-cell populations was aligned with a reduced count of EDU+ thymocytes and a simultaneous increase in cell cycle inhibitor Cdkn1a. These findings showed a decrease in CD4 and CD8 cell pools due to proliferation deficits rather than cell death [96].

One potential molecular pathway contributing to proliferation deficits in TrxR-1-deficient thymocytes is the dysregulation of cell cycle progression. TrxR-1 is known to play a role in redox signaling pathways that regulate cell cycle progression, including the modulation of cyclin-dependent kinase (CDK) activity and the expression of cell cycle inhibitors [23]. The disruption of TrxR-1-mediated redox signaling may lead to an aberrant activation of cell cycle inhibitors, such as Cdkn1a (p21), resulting in cell cycle arrest and impaired proliferation of CD4 and CD8 T-cells [97]. Furthermore, TrxR-1 deficiency may impact the expression or activity of transcription factors involved in T-cell development and proliferation. For example, factors such as T-cell factor 1 (TCF-1) and GATA-3 play critical roles in T-cell development and lineage commitment [98]. The dysregulation of these transcription factors in TrxR-1-deficient thymocytes may impair the differentiation and proliferation of CD4 and CD8 T-cells. Overall, the observed proliferation deficits in TrxR-1-deficient thymocytes likely involve a complex interplay of molecular pathways and factors, including the dysregulation of cell cycle progression, altered expression of transcription factors, and perturbation of signaling pathways, which are critical for T-cell activation and survival. Further investigations are needed to elucidate the specific mechanisms underlying these proliferation deficits and their impact on CD4 and CD8 T-cell populations in the context of TrxR-1 deficiency.

#### 3.3.5. Brain-Specific Trx-1-Null Mutation

Brain tissue is highly sensitive to redox balance and is therefore more vulnerable to oxidative stress compared to the liver or kidneys [99]. Due to higher levels of oxygen consumption, increased content of lipids with unsaturated fatty acids, elevated iron content, and lower activities of redox scavengers (e.g., superoxide dismutase, catalase, and glutathione peroxidase), brain metabolism is associated with increased ROS production [100]. Oxidative distress has been linked to several neurodegenerative disorders, such as amyotrophic lateral sclerosis, Alzheimer’s disease, Parkinson’s disease, multiple sclerosis, and Huntington’s disease [99,101,102]. The role of Trx-1 in neuronal homeostasis is extensively described in in vitro models [44,103]. However, to the best of our knowledge, no neuron/glial-specific Trx-1 knockdown animal model has been reported. A recent report has linked Trx-1 deficiency in the brain to epilepsy [104,105].

This study focused on a rare rat strain with frequent running seizures generated by the N-ethyl-N-nitrosourea (ENU) mutation [106]. To identify the potential mutation, animals with the desired phenotype were first backcrossed for more than 10 generations to achieve a higher frequency of this mutation in their offspring. Using genetic locus mapping methods, they identified a mutation in chromosome 5, which was then sequenced resulting in the identification of a single heterozygous missense mutation (phenylalanine replaced by leucine) in the exon region c.T160C of the Trx-1 gene. Intriguingly, this mutation was remote from the active site that was predicted to impair Trx-1 functions beyond dithiol/disulfide exchange. These rats presented with the highest frequency of seizures at 5 weeks of age. A histopathological assessment showed vacuolar degeneration in the midbrain (inferior colliculus and thalamus). Interestingly, no changes were observed in the hippocampus, which is the brain region responsible for the onset of epileptic activity in classic temporal lobe epilepsy [107]. Degeneration in the midbrain was also associated with decreased neuronal and oligodendrocyte count, myelin loss, astrogliosis, and increased expression of 8-OHd6 the marker of ROS-mediated DNA damage [107]. Trx-1 depletion also had an impact on energy metabolism in the thalamus, as shown by absolute quantitative values of NAD+, NADP+, UPD-glucose, etc. The dysregulation of NAD+ and NADP+ levels can further disrupt cellular energy production and redox homeostasis, leading to oxidative stress and impaired cellular function. Furthermore, changes in energy metabolism in the thalamus may impact neurotransmitter synthesis and signaling, synaptic function, and neural network activity, thereby contributing to the observed behavioral abnormalities and seizure activity in Trx-1-deficient animals. Disruptions in energy metabolism may impair neuronal excitability, synaptic transmission, and network synchronization, leading to hyperexcitability and seizure susceptibility.

Despite performing genetic mapping, the authors could not completely exclude other mutations in the gene resulting in this phenotype. Therefore, they used the CRISPR-Cas approach to reproduce the previously identified mutations in wildtype rats. The behavioral and histopathologic features in this knock-in model were similar to those found in previously studied mutants. Successful phenotype reproduction confirmed the major contribution of c.T160C mutation in the exon of the Trx-1 gene to this pathology. Interestingly, the severity of behavioral abnormalities was observed to decrease after a peak at 5 weeks, with a complete absence of seizure activity by week 9. Similarly, a histological assessment showed no signs of vacuolar degeneration, which was evident at 5 weeks. The nature of this spontaneous recovery remains to be investigated [104,105].

#### 3.3.6. Brain- and Neuron-Specific Knockouts of TrxR-1

The depletion of the Trx-1/TrxR-1 system has been linked to several neurodegenerative diseases, e.g., Alzheimer’s disease and Parkinson’s disease [102,108]. Defects in neural tube closure in embryos with constitutive TrxR-1 suggest the involvement of this antioxidant protein in the proper development of the neural system [69]. Soerensen et al. were the first to investigate the role of TrxR-1 in mice brains using a gene knockout approach [109]. This group used TrxR-1tm1Marc to generate two TrxR-1 knockout strains: (1) by crossing with Nestin-Cre mice, which drove Cre-recombinase to precursor cells of both neuronal and glial lineages; (2) by crossing with Tα1-Cre mice, which drove Cre-recombinase specifically to neurons [105]. Interestingly, these two mutant strains displayed drastically different phenotypes. Mice with TrxR-1 depletion in neuronal and glial lineages had significant growth retardation, reaching only 50% of their estimated body mass compared to controls by 4 weeks. Additionally, they displayed notable impairments in coordinated movement (e.g., ataxic gate, balance problems, and tremor), suggesting primary involvement of the cerebellum. Histologically, these mice displayed striking cerebellar hypoplasia with reduced proliferation in the external granular layer. Additionally, a strong reduction of fissure formation and laminar organization of the cerebellar cortex was present in this mutant strain. On the cellular level, Soerensen et al. observed disruption of the Bergmann glia network along with ectopic localization and reduced arborization of Purkinje cells. According to the authors, a reduction in size of the cerebellum and decreased cell count were not associated with apoptosis, but rather with a reduced proliferation of granule cell precursors in the external granule layer [109].

In contrast, the mice with a neuron-specific ablation of TrxR-1 did not present any significant phenotypes. They showed no significant behavioral or histological abnormalities [109]. Taken together, these findings may suggest that TrxR-1 expression has a greater impact on neural precursors and glial cells rather than on mature neurons. One potential explanation for the observed phenotypic variations between these two mutant strains lies in the differential roles of TrxR-1 in neural precursors, glial cells, and mature neurons. TrxR-1 is known to play critical roles in maintaining redox homeostasis, regulating cellular proliferation and differentiation, and protecting cells from oxidative stress [23]. Therefore, disruptions in TrxR-1 expression may have differential effects on these cellular processes, depending on the cell type and developmental stage.

In the mutant strain with a TrxR-1 depletion in neuronal and glial lineages, the observed growth retardation and cerebellar hypoplasia suggest that there were impairments in neural precursor proliferation and differentiation. TrxR-1 may play a crucial role in regulating the proliferation of neural precursor cells in the external granular layer of the cerebellum, as evidenced by the reduced proliferation observed in this mutant strain [109]. Additionally, the disruption of the Bergmann glia network and aberrant localization of Purkinje cells further underscore the importance of TrxR-1 in regulating glial cell development and neural circuit formation during cerebellar development [109]. In contrast, the absence of significant phenotypic abnormalities in mice with a neuron-specific ablation of TrxR-1 suggests that TrxR-1 expression in mature neurons may have a less pronounced impact on overall neural function [109]. This can be due to higher dependence of mature neurons on alternative redox regulatory mechanisms rather than TrxR-1, compared to neural precursor cells and glial cells. Overall, the observed phenotypic variations between these two mutant strains may be attributed to the differential roles of TrxR-1 in regulating cellular processes such as proliferation, differentiation, and the oxidative stress response in neural precursors, glial cells, and mature neurons. Further investigations into the specific molecular mechanisms and pathways underlying these phenotypic differences are warranted to elucidate the role of TrxR-1 in neuronal and glial development and function.

## 4. Concluding Remarks

The generation of animal strains lacking Trx-1 or TrxR-1 activity has proven invaluable for exploring the roles of this redox system in maintaining cellular homeostasis. Despite a limited number of characterized animal models, the available literature underscore the indispensable role of the Trx-1 system in the brain and heart. However, the liver- or pancreas-specific deletion of these genes seems to be normal, with an overall healthy life span for the animals (Figure 2). This might reflect the differential dependency of various cell types to different antioxidants. For example, a higher dependency of rat brain mitochondria on the Trx system for peroxide scavenging has been shown by the addition of specific Trx and GSH inhibitors [110]. These observations suggest that the Trx-1/TrxR-1 system’s specific functions in different organs likely depend on organ-specific roles. While it exhibits a general cytoprotective effect across various cell types, it also influences cell proliferation and differentiation in mitotic cells. In post-mitotic cells, the Trx-1/TrxR-1 system predominantly has an antiapoptotic effect. However, in cell types with high metabolic activities, such as hepatocytes or cardiomyocytes, it is also involved in regulating mitochondrial metabolism. It is also plausible that cells at different stages will respond differently to stress conditions and require differential involvement of cellular antioxidants.

Although, theoretically, the effects of Trx-1 and TrxR-1 knockouts are expected to be identical, experimental models of Trx-1 deficiency tend to have more pronounced effects compared to TrxR-1 deficiency. Studies with hepatocyte Trx-1 vs. TrxR-1 knockout models showed that Trx-1 plays other roles than simply being a substrate of TrxR-1. In fact, Trx-1 has been shown to regulate the activity of many other proteins in the cytoplasm and nucleus. It is therefore expected that a functional deletion of Trx-1 will have more devastating effects than TrxR-1 depletion. While there is substantial literature on the Trx system in the heart, we have just started to understand the impact of the Trx system on the brain. Novel Trx-1 deletion models are currently available that will highlight the multifaceted functions of Trx-1, which extend beyond its antioxidant capacity. The summarized characterization of the discussed Trx-1 and TrxR-1 knockout mice strains is provided in Table 1.

## Figures and Tables

**Figure 1 biology-13-00180-f001:**
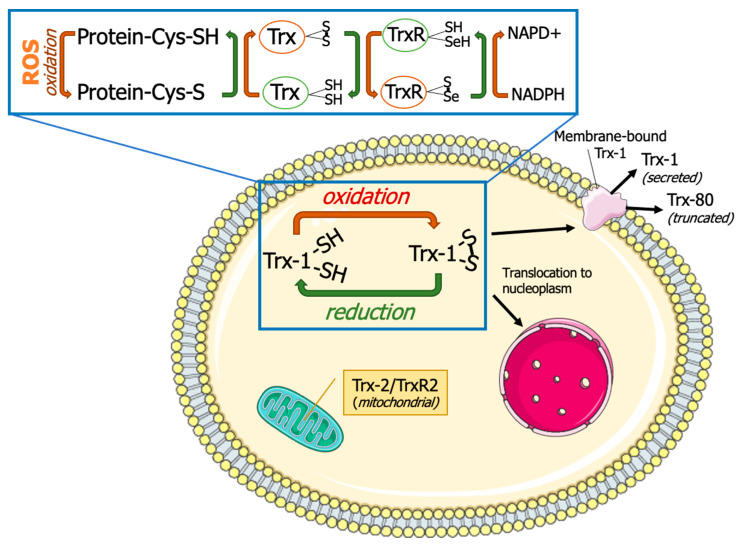
**Thioredoxin system**. Thioredoxins (Trxs) donate hydrogen to resolve disulfide bonds (S-S) in protein cysteines (Cys). Oxidized Trx is reduced by thioredoxin reductases (TrxRs) with the help of reduced nicotinamide adenine dinucleotide phosphate (NADPH). Thioredoxin-1 (Trx-1) is typically localized in cytoplasm but can also undergo nuclear translocation, membrane binding, or secretion. Thioredoxin-2 (Trx-2) and thioredoxin reductase-2 (TrxR-2) are localized in mitochondria. This figure was partly generated using Servier Medical Art, provided by Servier, licensed under a Creative Commons Attribution 3.0 unported license (https://creativecommons.org/licenses/by/3.0/) (accessed on 15 January 2024).

**Figure 2 biology-13-00180-f002:**
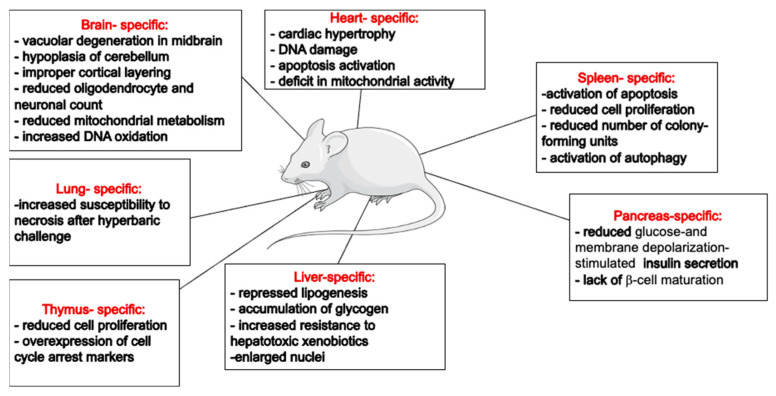
**Summary of reported organ-specific effects of Trx system’s downregulation/knockouts in mice.** This figure was partly generated using Servier Medical Art, provided by Servier, licensed under a Creative Commons Attribution 3.0 unported license (https://creativecommons.org/licenses/by/3.0/).

**Table 1 biology-13-00180-t001:** Characterization of Trx isoforms.

	Trx-1	Trx-2
Tissue specificity	Ubiquitous	Ubiquitous
Subcellular localization	Cytoplasm Nucleus Extracellular matrix	Mitochondria
	thioredoxin-1	thioredoxin-2
Protein Sequence (human)	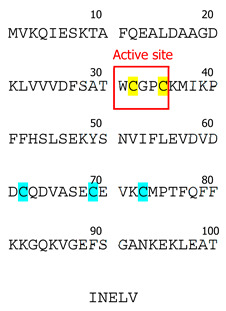	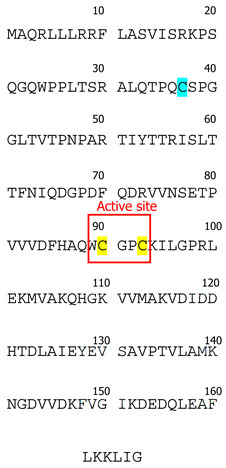

Cysteines located within the active site of thioredoxin-1 (Trx-1), thioredoxin-2 (Trx-2), and thioredoxin-3 (Trx-3) are highlighted with yellow. Cysteines outside of the active site are highlighted with turquoise.

**Table 2 biology-13-00180-t002:** The roles of Trx-1 in cellular processes in vitro.

Subcellular Localization	Cell Function	Target	Cell Type	References
**Cytoplasm**	** *Antioxidative defense* **	Disulfide-containing proteins	All	Bertini et al. [21], Ueno et al. [37], Liu et al. [38], Hwang et al. [39], Akterin et al. [40], Nakamura et al. [41], Tao et al. [28], Arner et al. [23]
	** *Apoptosis* **	ASK-1	Mv1Lu (Mink Lung Epithelial Cells), L929 (mouse fibroblast cell line) and 293 (mouse fibroblast cell line)	Saitoh et al. [42]
		MST-1	MEF (murine embryonic fibroblasts)	Chae et al. [43]
		Casp-6	SH-SY5Y (human neuroblastoma cells)	Islam et al. [44]
		Casp-3	Jurkat (immortalized human T-lymphocytes)	Mitchell et al. [45]
	** *Autophagy* **	mTOR	Rat cardiomyocytes	Oka et al. [46]
		ATG7	Rat cardiomyocytes	Nagarajan et al. [47]
		ATG4	Yeast Saccharomyces cerevisiae	Prez-Perez et al. [48]
		LC III-B	HLE-B3 (human lens epithelial cells)	Hu et al. [49]
	** *Cytoskeleton organization* **	Actin	SH-SY5Y (human neuroblastoma cells)	Wang et al. [50]
		Tubulin	Purified porcine tubulin	Landino et al. [51]
		CRMP-2	Embryonic DRG neurons from Sprague Dawley rats	Morinaka et al. [52]
		NGF	PC-12 (adrenal phaeochromocytoma cells)	Bai et al. [53]
	** *Signaling transduction* **	AKT and PTEN	Neuroblastic neoplasms, and neuroblastoma cell lines	Sartelet et al. [54]
	** *Inflammation* **	Monocyte chemoattractant protein-1 (MCP-1)	Trx-1 Overexpressing/knockdown EA.hy 926 and bovine aortic endothelial cells	Chen et al. [55]
	** *Immunomodulatory* **	Regulatory T-cells	Tregs (regulatory T-lymphocytes)	Mougiakakos et al. [19]
**Nucleus**	** *Gene regulation* **	NFk-B, AP-1, and Ref-1	L929 (mouse fibroblast cell line), HeLa (cervical cancer cells), COS-7 (African green monkey kidney fibroblast-like cells)	Chen et al. [55] Schenk et al. [56]
		HIF-1a	HeLa (cervical cancer cells)	Naranjo-Suarez et al. [57]
		Oct-4	Embryonic stem cells	Guo et al. [58]
		HDAC4	Cardiac myocytes	Ago et al. [59]
	** *DNA binding* **	NRF-2	HeLa (cervical cancer cells)	Hansen et al. [60]
	** *Transportation to nucleus* **	Glucocorticoid receptor	COS7 and CV-1 (African green monkey kidney fibroblast-like cells), HeLa (cervical cancer cells)	Makino et al. [61]
		Estrogen receptor	MCF-7 (breast cancer cells)	Rao et al. [62]
**Cell membrane**	** *Immunomodulatory* **	Complement deposition	HUVEC (human umbilical vein endothelial cells)	King et al. [63]
**Extracellular matrix**	** *Chemokine-like* **	Chemoattraction	human monocytes, PMNs, leucocytes	Bertini et al. [21] Nordberg et al. [64]

## Data Availability

Not applicable.

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
