# Peer review of "Insights into the Multifaceted Roles of Thioredoxin-1 System: Exploring Knockout Murine Models"

_biology, 2024, doi:10.3390/biology13030180_

Round 1

Reviewer 1 Report

Comments and Suggestions for Authors

Dear Editors,

In this manuscript, Tetiana et al. offer a preliminary review emphasizing insights gained from Thioredoxin-1 knockout mouse models. The well-crafted review predominantly revolves around Thioredoxin-1 system transgenic models, incorporating both Thioredoxin and Thioredoxin reductase proteins. In light of the content, a modification to the title for increased precision is necessary. Moreover, some terms are not accurate, e. g. TXNIP is thioredoxin interacting protein; “thioredoxin family” may be generally-used “thioredoxin system”, because the thioredoxin family is easy to be confused with “thioredoxin superfamily” which normally indicates the proteins with “thioredoxin fold”. The figure number is not enough for a comprehensive understanding for the topic, more figures are required. Additionally, the following points to enhance the overall quality of the review should be made before the acceptance of publication.

Title Revision: Consider refining the title to better reflect the manuscript's primary focus on Thioredoxin-1 knockout mouse models and the associated implications for understanding the Thioredoxin-1 system. This adjustment will provide a more accurate representation of the review.

Major comments:

1) The order of the headings should be changed to:

1.      Introduction

2.      Trx system enzymes

2.1.   Isoforms

2.2.   Roles of Trx1 (Just before this sentence “Trx1 is a structurally and functionally unique protein with multiple roles in cellular homeostasis…….”)

3.      As such ….

2) “In this review, we aim to summarize the molecular, morphological, and functional characteristics of animal models with abolished Trx1/TrxR1 system in various organs. By comparing the described effects of Trx1/TrxR1 loss in organ- and cell-type-specific knock-outs, we aimed to dissect the common functions of Trx1/TrxR1 system from those restricted to particular organ or cell type. The present review provides a valuable resource for those researchers who investigate tissue-specific roles of the Trx1 system or aim to generate specific Trx1/TrxR1 knockout strains.”

This paragraph should conclude the introduction part (with some modification to include the “Trx family proteins” part), followed directly with (2. Trx family proteins). Example: In this review, we aim to discuss the TRX family proteins and mammalian isoforms, while also exploring the roles of Trx1. Additionally,…etc., If you can also expand one or two paragraphs in the introduction to give a general idea on the whole section that will be better.

3) “Trx1 and Trx2 are the most researched isoforms of this family. We will first briefly discuss Trx2 and then will focus majorly on Trx1.”  This sentence should either be combined with the previous or later paragraph.

2. Review of murine Trx1 system knockout models

4) It is essential to provide clarity on three distinct categories of knockout models for the readers. These include models characterized by Constitutive Trx1 and TrxR1 knockout, those featuring General inducible Trx1 knockout and dominant negative Trx1, and models specifically designed for Organ-specific Trx1/TrxR1 knockouts. Additionally, it is important to explain the differences among these models, outline their practical applications, and justify the reasons for their utilization. And discuss the advantages and limitations for each. For instance, in the case of Constitutive Trx1 and TrxR1 knockout, it would be beneficial to introduce the model and explain its significance and applications.

2.1. Constitutive Trx1 and TrxR1 knockout

5) Can the authors provide additional clarification on the mechanism through which Trx1 deficiency leads to deficits in trophoblast formation, particularly in relation to the observed defects in DNA synthesis?

6) Considering the varied impacts on organ development, what insights can be drawn from the fact that heart development in TrxR1-deficient embryos showed no significant deviations from controls?

7) Given the association between TrxR1 deficiency and disruptions in differentiation rather than proliferation, are there specific markers or pathways implicated in these processes that the authors could further explore or discuss?

2.2. General inducible Trx1 knockout and dominant negative Trx1

8) Could you elaborate on the rationale behind choosing the dominant negative form of Trx1 as an alternative approach to disrupt the activity of wild-type Trx1, and how does it compare to the Tamoxifen-inducible model in terms of implications for studying Trx1 function?

2.3. Organ-specific Trx1/TrxR1 knockouts

2.3.1. Heart-specific

9) How do the findings from the cardiac-specific Trx1 functional depletion model (using α-myosin heavy chain-Cre mice strain) compare to those from the Trx1 heart-specific knockout model (achieved by breeding Trx1fl/fl animals with Myh6-Cre mice)? Are there distinct differences in the phenotypic outcomes and molecular responses between these models?

2.3.2. Liver-specific Trx1:

10) The study reports that liver-specific downregulation of Trx1 and TrxR1 is not lethal, with only a 15% lethality rate and fertile adult mice. Could the authors discuss the potential compensatory mechanisms that allow for the survival of these mice despite the downregulation of critical redox regulators in hepatocytes?

11) The study highlights the repression of lipogenic genes and glycogen accumulation in TrxR1-deficient livers, along with enhanced detoxification mechanisms. Could the authors discuss the potential implications of these changes on overall hepatic function and metabolism?

2.3.3. Pancreatic β-cell:

12) The study reports no significant difference in glucose tolerance and plasma insulin levels in mutant mice after a 4-hour fasting period. Could the authors discuss the potential reasons for these observations and whether compensatory mechanisms might be at play?

13) Given that the reduction in β-cell function is attributed to disturbances in the maturation of TrxR1-deficient β-cells, could the authors discuss the significance of identifying factors involved in β-cell maturation and their impact on glucose sensing? Could the authors speculate on the potential molecular mechanisms or factors that might be involved in this maturation process and how they relate to the regulation of glucose sensing?

2.3.4. T-Cell-specific:

14) Given the reported reduction of Txnip in activated T-cells and the upregulation of Trx1 and TrxR1, could the authors discuss the potential compensatory mechanisms that might exist in the absence of TrxR1 in T-cells?

15) The study suggests that the decrease in CD4 and CD8 cell pools in TrxR1-deficient thymuses is due to proliferation deficits rather than cell death. Could the authors discuss the specific molecular pathways or factors involved in the observed proliferation deficits and how they relate to the regulation of CD4 and CD8 T-cell populations?

2.3.5. Brain-specific:

16) The impact of Trx1 depletion on energy metabolism in the thalamus is mentioned. Could the authors discuss the implications of these changes in energy metabolism and how they might contribute to the observed pathology?

2.3.6. Brain- and neuron-specific:

17) The study reports significantly different phenotypes in mice with TrxR1 depletion in neuronal and glial lineages compared to those with neuron-specific ablation of TrxR1. Could the authors discuss potential molecular mechanisms or pathways that might explain the observed phenotypic variations between these two mutant strains, particularly focusing on the role of TrxR1 in neural precursors, glial cells, and mature neurons?

Minor comments:

18) In the second line in the introduction

H2O2 should be H2O2, the author should revise the text thoroughly to avoid such mistakes.

19) There are a few of mistakes in organizing the headings of the review, you can revise them to be identical in placement.

Reviewer 2 Report

Comments and Suggestions for Authors

In this manuscript, Shcholok and Eftekharpour describe the different biological effects of various thioredoxin and thioredoxin-reductase knockout and transgenic mice, after going through some fundamental thioredoxin biochemistry.  In general, the manuscript is informative and will be of use to those in the field.  My comments are mainly stylistic, but hopefully they will help the authors revise the manuscript.  With some minor edits and better integration of ideas, the manuscript should be ready for publication.  

Major comments

Particularly in the second half of the document, there is limited integration of the information described.  It seems like each paragraph in the second half of the document just summarizes a single paper (or two or three, at most).  Each paragraph describes the effect of that particular knockout in a specific tissue-type, rather than describing the unifying themes that emerge from studying the knockouts as a whole.  Thus, the information, while organized, is a little disjointed, dry, and overly descriptive.  The authors have done a nice job of summarizing the information, but rewriting it to be more seamless/fluid might improve readability.  

Some sections could use more citations. In particular, S-nitrosation of Trx1 on the non-active site cysteines has been studied by the Marletta/Tannenbaum and Holmgren labs, but their papers are not cited.  Likewise, oxidation of the non-active site cysteines in Trx1 has been reported by the lab of Dean Jones.  In addition, there is quite a bit of information about the secreted forms of Trx1 functioning in the ECM to contribute to celiac disease (work from the Khosla lab and potentially other lab); this work is not cited.  

Minor comments

•In the abstract, it would be more accurate to say that thioredoxin is involved in ‘protein disulfide reduction’ rather than ‘cysteine reduction.’  

•Above Figure 1, TXNIP is ‘thioredoxin interacting protein,’ not ‘thioredoxin inhibitory protein,’ to my knowledge.

•In Figure 1, the disulfide bonds formed in a Trx substrate and in Trx1 itself are not visible in the top panel or in the cellular diagram.  Thus, this figure could be more chemically accurate.  

•I might avoid mentioning Trx3 (in the sentences above Table 1) as this area seems a bit shaky.  There has not been much progress in this area for the past 20 years to my knowledge.  Instead, it might be better to focus exclusively on Trx1 and Trx2.  

•In Table 1, the terminology should be ‘active site’ instead of ‘active centre/center’ as shown in the table itself and in the accompanying legend.  Also, the cyan highlights of the non-active site cysteines are illegible (black text within a blue box), so picking a different color to depict would be useful. 

•On page 4, 1st full paragraph – omit ‘and tissues’ from the opening sentences, as tissues do not have a cytoplasm, per se.  

•On page 4, 3rd full paragraph – there is mention of (Packer 1995).  This reference should be cited numerically to be consistent with the other citations in the manuscript.

•On the last line of page 4, change ‘vital’ to ‘various,’ as some Trx1 substrates do not carry out ‘vital’ roles.   

•In Table 2, specific substrates of Trx are listed as Trx targets in some cases, whereas in other cases, Trx ‘targets’ include ‘regulatory T cells,’ ‘complement deposition,’ and ‘chemoattraction.’  None of the latter examples are proteins, so the examples provided are inconsistent with the molecular function of thioredoxin described elsewhere in the table (where specific proteins are listed).  Likewise, in the first line of the paragraph ‘oxidized proteins’ should read ‘disulfide-containing proteins,’ as there are other forms of protein oxidation besides disulfide-linked cysteines.  

•At the bottom of page 7 and the top of page 8, the term ‘dominant negative’ is confusing and could be misleading.  I would use the term ‘catalytically inactive’ instead.  

•In the next to last sentence of the paragraph that begins page 8, what form of Trx1 is overexpressed in the ‘Trx1 overexpressing mice?’ I’m assuming it is the wild-type protein, but it would be good to state explicitly.

•On page 9, section 2.3.2, liver-specific ‘knockout’ or ‘deletion’ rather than ‘downregulation’ – ‘Downregulation’ should be reserved for genetic knockdowns, rather than knockouts.  

•On page 9, last paragraph:  “lipogenic genes, reflected in” rather than ‘lipogenic genes reflecting in…’  

Comments on the Quality of English Language

Overall the English is good.  Some sentences could use some edits with comma or semicolon placement (i.e., punctuation is missing).

Round 2

Reviewer 1 Report

Comments and Suggestions for Authors

I only have a small suggestions and comments to add to improve the quality of this review.

1)In this review, we provide an overview of murine Trx1 system knockout models generated using three distinct genetic modification techniques. These techniques include constitutive gene knockout, inducible gene knockout, catalytically inactive gene knock-in, and organ-specific gene knockout. …… application, advantages, and disadvantages, we recommend consulting our recent review paper [65].”

Consider combining the small paragraphs to make only two paragraphs.

2) It is essential to provide clarity on three distinct categories of knockout models for the readers. These include models characterized by Constitutive Trx1 and TrxR1 knockout, those featuring General inducible Trx1 knockout and dominant negative Trx1, and models specifically designed for Organ-specific Trx1/TrxR1 knockouts. Additionally, it is important to explain the differences among these models, outline their practical applications, and justify the reasons for their utilization. And discuss the advantages and limitations for each. For instance, in the case of Constitutive Trx1 and TrxR1 knockout, it would be beneficial to introduce the model and explain its significance and applications.

What I mean here is to start each subsection by giving a general background about the model itself, definition, advantages, disadvantages application, then follow into the studies and literature review, so it can be easier to understand the paper for the reader. But this hasn’t been done clearly, I hope that the can consider it.

3) “Global inborn Trx1 deletion is reported to be embryonically lethal [1]. While the ……. determination and differentiation during embryonic development) [66].”

Consider breaking it down to two paragraphs, it’s a bit long, and focus on the references.

4) It is crucial to support the information provided with relevant references to enhance the credibility and traceability of the scientific claims made in the text. For instance:

Some paragraphs or phrases that must contain references are:

“They noted that TrxR1 deficient livers exhibit repression…. Therefore, these modifications in hepatocyte metabolism resulted in the preconditioning of mutant hepatocytes to even more robust elimination of xenobiotics compared to wild-type littermates.”

“The changes in hepatic function and metabolism have broad implications for overall liver health and susceptibility to environmental toxins and drug-induced liver injury…… thiol reducing systems.”

“Degeneration in the midbrain was also associated with decreased neuronal and ……transmission, and network synchronization, leading to hyperexcitability and seizure susceptibility.”

“In contrast, the absence of significant phenotypic abnormalities in mice with neuron specific ablation of TrxR1 suggests that TrxR1 expression in mature neurons may have a less pronounced impact on overall neural function. Mature neurons may rely on alternative redox regulatory mechanisms or may be less susceptible to TrxR1 depletion compared to neural precursor cells and glial cells.” Combine it with the previous paragraph. And the same goes for “Overall …. function”

5) “One potential molecular pathway contributing to proliferation deficits in TrxR1 deficient thymocytes is the dysregulation of cell cycle progression. TrxR1 is known to play a role in redox signaling pathways that regulate cell cycle progression, including modulation of cyclin-dependent kinase (CDK) activity and expression of cell cycle inhibitors [22]. Disruption of TrxR1-mediated redox signaling may lead to aberrant activation of cell cycle 562 inhibitors, such as Cdkn1a (p21), resulting in cell cycle arrest and impaired proliferation of CD4 and CD8 T-cells [94].

Furthermore, TrxR1 deficiency may impact the expression or activity of transcription factors involved in T-cell development and proliferation. For example, factors such as T-cell factor 1 (TCF-1) and GATA-3 play critical roles in T-cell development and lineage commitment [107]. Dysregulation of these transcription factors in TrxR1-deficient thymocytes may impair the differentiation and proliferation of CD4 and CD8 T-cells.

Overall, the observed proliferation deficits in TrxR1-deficient thymocytes likely involve a complex interplay of molecular pathways and factors, including dysregulation of cell cycle progression, altered expression of transcription factors, and perturbation of signaling pathways critical for T-cell activation and survival. Further investigation is needed to elucidate the specific mechanisms underlying these proliferation deficits and their impact on CD4 and CD8 T-cell populations in the context of TrxR1 deficiency.”

Combine in one paragraph. No need to make them three separate phrases.

Author Response

We sincerely appreciate the constructive feedback provided by the reviewers, and we have carefully addressed each point raised to enhance the quality and clarity of our manuscript.

Comments 1, 3-5: Thank you for your feedback. We have addressed your suggestions by revising the subsections to provide clearer background information on knockout models, combining paragraphs for readability, and ensuring references support the information provided.

Comment 2: Thank you for your insightful comments and suggestions regarding the categorization and characterization of knockout models in our review paper. We acknowledge the importance of providing clarity on three distinct categories of knockout models, as highlighted in your comment. It is important to note that the contexts of model creation varied significantly among these cases, leading to different approaches to model characterization. As a result, the amount of information available for each model may vary. We have made efforts to reiterate what the authors reported in order to provide maximum information for our readers. While we acknowledge that a standardized approach to model characterization would enhance clarity, we believe that presenting the information as reported by the original authors offers transparency and accuracy regarding the characteristics and applications of each model. We hope this explanation clarifies the approach we've taken and assures you of our commitment to enhancing the paper's clarity and accessibility for our readers.

Warm regards